# Prognostic Significance of STING Immunoexpression in Relation to HPV16 Infection in Patients with Squamous Cell Carcinomas of Oral Cavity and Oropharynx

**DOI:** 10.3390/biomedicines10102538

**Published:** 2022-10-12

**Authors:** Beata Biesaga, Ryszard Smolarczyk, Anna Mucha-Małecka, Justyna Czapla, Janusz Ryś, Krzysztof Małecki

**Affiliations:** 1Department of Tumor Pathology, Maria Sklodowska-Curie National Research Institute of Oncology, Cracow Branch, Garncarska 11, 31-115 Cracow, Poland; 2Center for Translational Research and Molecular Biology of Cancer, Maria Sklodowska-Curie National Research Institute of Oncology, Gliwice Branch, Wybrzeże Armii Krajowej 15, 44-102 Gliwice, Poland; 3Department of Radiotherapy, Maria Sklodowska-Curie National Research Institute of Oncology, Cracow Branch, Garncarska 11, 31-115 Cracow, Poland; 4Department of Radiotherapy for Children and Adults, University Children’s Hospital of Krakow, Wielicka 265, 30-663 Cracow, Poland

**Keywords:** head and neck cancers, prognostication, HPV16 infection, STING immunoexpression

## Abstract

Infection with HPV16 in cancers of the oral cavity (OCSCC) and oropharynx (OPSCC) is, today, an important etiological and prognostic factor. Patients with HPV-positive OPSCC have a better prognosis than uninfected patients. However, in over 40% of these patients, cancer progression is noticed. Their identification is particularly important due to the ongoing clinical trials regarding the possibility of de-escalation of anticancer treatment in patients with HPV-positive OPSCC. Some studies suggest that there is possibility to differentiate prognosis of HPV16-positive patients by STING (Stimulator of Interferon Genes) immunoexpression. The aim of the present study was to analyze the influence of STING immunoexpression on overall (OS) and disease-free survival (DFS) of patients with HPV16-positive and -negative OCSCC and OPSCC. The study was performed in a group of 87 patients with OCSCC and OPSCC for which in our earlier study active HPV16 infection was assessed by P16 expression followed by HPV DNA detection. To analyze STING immunoexpression in tumor area (THS) and in adjacent stromal tissues (SHS) H score (HS) was applied. In the subgroup with HPV16, active infection patients with tumors with THS had significantly better DFS (*p* = 0.047) than those without THS. In this subgroup, TSH did not significantly influence OS, and SHS did not significantly correlate with OS and DFS. In the subgroup of patients without active HPV16 infection, THS and SHS also did not significantly influence patients’ survival. Presented results indicated prognostic potential of tumor STING immunoexpression in patients with active HPV16 infection in cancers of oral cavity and oropharynx.

## 1. Introduction

Recently, human papillomavirus (HPV) infection has been the cause of a growing number of squamous cell carcinomas of the head and neck (HNSCCs), especially within the oropharynx (OPSCC) [1]. The most frequently detected virus type is HPV16. HPV-dependent HNSCCs differ significantly from HPV-negative ones (most often developing as a result of exposure alcohol and tobacco) in terms of epidemiologic, clinical and histopathological features. Numerous clinical studies have also revealed that HPV-positive HNSCC patients have a better prognosis than uninfected patients. This observation was confirmed by the results of four meta-analyses covering about 200 studies [2,3,4,5]. However, as shown in the hazard ratios from these meta-analyses, in the subgroup of HPV-positive HNSCC patients, cancer progression occurs in over 40% of these patients. Their identification is particularly important due to the ongoing clinical trials regarding the possibility of de-escalation of anticancer treatment in HPV-positive patients with OPSCC [6]. Therefore, there is an urgent need to indicate new prognostic and/or predictive factors, allowing for the identification of a subgroup of patients with HPV infection that benefit from the de-escalation of treatment.

One of the possibilities of treatment de-escalation in the case of HPV positivity is the use of a combination of a STING agonist plus anti-CTLA-4 or anti-PD1 ICB treatments. This strategy is actually tested in ongoing clinical trials (NCT02675439, NCT03172936, and NCT03010176) [7]. The STING (STimulator of INterferon Genes) is a transmembrane endoplasmic reticulum (ER) protein, which plays a key role in the cell’s response to the presence of DNA in the cytoplasm [8]. Briefly, cytoplasmic DNA is recognized by the enzyme cGAS, inducing production of the cyclic dinucleotide 2′,3′-cGAMP, which binds to STING, causing dimerization of this protein and at the same time its activation. Activated STING migrates to a perinuclear Golgi-like compartment, where oligomerizes to recruit and activate TANK-binding kinase 1 to phosphorylate the transcription factor IRF3, stimulating an IFN response.

The influence of HPV16 infection on STING expression is not fully explained. In the experimental studies performed on cancer cell lines, it was shown that E7 oncoprotein of HPV16 blocked cGAS-STING response in infected cells [9]. It was also found that CRISP/Cas9-mediated loss of E7 restored STING response [10]. These findings suggest mechanisms for silencing the innate immune response by viruses. On the other hand, some authors have noticed STING expression in HPV-related HNSCC and its lack in HPV-negative HNSCC [11,12,13], which may suggest prognostic potential in patients with HPV-related HNSCC. However, it should be noted that, according to our best knowledge, prognostic significance of STING immunoexpression has never been assessed in a group of patients with OCSCC and OPSCC in relation to HPV infection.

In our earlier study, among 87 patients with OCSCC and OPSCC we found HPV16 transcriptionally active infection (P16 overexpression and positivity of HPV DNA in quantitative polymerase chain reaction—qPCR) in, respectively, 16.0 and 37.1%; in laryngeal and hypopharyngeal cancers, these percentages were significantly lower [14]. Therefore, in the light of above-mentioned unclear results concerning the mutual relation between HPV presence and STING expression, we decided to perform a translational study, aimed at analyzing the influence of STING immunoexpression on overall survival (OS) and disease-free survival (DFS) of patients with HPV16-positive and HPV16-negative OCSCC and OPSCC.

## 2. Materials and Methods

### 2.1. Study Design

The study was a retrospective analysis performed in a group of 77 patients with squamous cell carcinoma of the oral cavity and oropharynx who were treated between 2007 and 2014 in Maria Skłodowska-Curie Memorial Cancer Center and Institute of Oncology, Cracow Branch (Figure 1). The inclusion criteria were as follows: (1) squamous cell carcinoma of oral cavity and oropharynx, (2) no distant metastasis at the moment of diagnosis, (3) assessment of active HPV16 infection in our earlier study [14], and (4) formalin-fixed and paraffin-embedded blocks with a sufficient amount of cancer tissue for immunohistochemistry. For our previous research, all FFPE underwent histological reverification in order to confirm tumor histology (squamous cell carcinoma), histologic grade and degree of keratinization [14]. Pathologists also selected paraffin blocks in which the tumor component covered > 50% of the slide area.

### 2.2. Immunohistochemistry

IHC staining was performed in typical FFPE sections. Deparaffinization and rehydration of sections were followed by an antigen-unmasking procedure (heating of slides in citrate buffer (10 mM sodium citrate, 0.05% Tween 20, pH 6.0)) in the microwave (850 W) for 20 min) and quenching of endogenous peroxidases (30 min incubation in 0.3% hydrogen peroxide, 37 °C). Next, 90 min incubation with diluted (1:100) primary antibody (STING mAb (D2P2F), Cell Signaling Technology, Danvers, MA, USA) in 37 °C was carried out. The reaction was visualized using BrightVision system (Immunologic, Duiven, The Netherlands) and 0.01% 3.3-diaminobenzidine tetrahydrochloride (Vector Laboratories, Inc., Burlingame, CA, USA). The slides were counterstained with Mayer’s hematoxylin. For negative control, phosphate-buffered saline (PBS) was substituted for each primary antibody. Positive control includes cervical cancer exhibiting high expression of Nanog DAB as a chromogenic substrate. Sections incubated with the phosphate buffer instead of the primary antibody served as a negative control. To each series of staining positive control was added, which included SCC of tongue exhibiting high immunoexpression of STING.

All evaluations were performed blinded to the study endpoint. Similar to other authors [13], H score (HS) was applied to analyze the intensity of STING expression in the tumor area (THS) and in adjacent stromal tissues (SHS) (Figure 2a–d). This score includes the intensity of the staining and the number of positive stained cells. HS was calculated according to the formula: H-score = (1 × percentage of weakly positive cells) + (2 × percentage of moderately positive cells) + (3 × percentage of strongly positive cells), giving a range from 0 to 300. The cut-off point for STING expression/its lack was selected based on the minimal *p* value method, described in detail in Section 2.3.

### 2.3. Statistical Analysis

Descriptive statistics were used to determine mean and median values of continuous variables and standard errors of means (SE). Student’s *t*-test was applied to establish the significance of differences between means. Associations between categorical variables were analyzed using Pearson c2 test. To analyze the prognostic potential, two endpoints were adopted: 5-year overall survival (time from the end of therapy until death from any cause) and disease-free survival (time from the end of therapy until the first documented evidence of recurrent disease—treatment failure, locoregional recurrence, distant metastasis). Survival curves were calculated using Kaplan–Meier estimates, and differences between groups were tested by the log-rank test. Univariate and multivariate survival analyses were carried out according to the Cox proportional hazards model (forward stepwise procedure). All statistical tests were two-sided, and *p* < 0.05 was considered significant. Statistical analyses were carried out using the Statistica v.13.0 program (StatSoft, Tulsa, OK, USA).

## 3. Results

### 3.1. Patients

In the group of 77 patients included in the study, there were 22 with cancers localized in the oral cavity (28.6%) and 55 (71.4%) with cancers in the oropharynx cancer. In the analyzed group, men prevailed (*n* = 57; 74.0%). The mean age of 77 patients was 57.7 ± 1.1 (SE), with a median value 59 years. In this group, four patients (5.2%) had tumors in clinical stage II, 17 (22.1%) in stage III and 56 (72.7%) in stage IV. Active HPV16 infection was noticed in 24 patients (32.9%) [14]. The detailed epidemiological, clinical and histopathological features are presented in Table 1.

Among 77 patients, 30 (39.0%) were treated with postoperative radiotherapy (RT) or RT alone. Total dose of RT ranged 20.0–66.0 Gy, with a mean value of 59.1 Gy ± 2.6, number of fractions: 5–40 and fraction dose: 1.8–4.0 Gy. CRT-CisPt as a definitive treatment or as an adjuvant treatment after surgery was applied for 28 patients (36.3%). In this subgroup, the total dose of RT was in the range 28–70 Gy (mean value: 64.1 Gy ± 1.6), applied in 14–35 daily fractions of 2.0–2.2 Gy. During RT, cisplatin (CisPt) was administrated according to two schemes: (1) 100 mg CisPt/m^2^ every third week of RT in two to three courses or (2) 40 mg CisPt/m^2^ every week of RT in three to six courses (depending on the patient’s condition and the severity of early normal tissue reactions). In turn, 19 patients (24.7%) were treated with induction chemotherapy (CisPt: +5-fluorouracil + taxanes), followed by RT (total dose: 28–70 Gy, mean value: 64.7 Gy ± 2.7; fraction dose: 2.0 Gy, number of doses: 14–25).

In the group of 77 patients, 37 patients (67.9%) were alive at the time of the study, 22 (17.4%) died from cancer disease and 18 (14.7%) died from other reasons, mainly from cardiovascular disease. Regression of cancer disease was noticed in 51 persons (65.1%), and tumor progression was observed in 26 patients (34.9%; treatment failure = 5, locoregional recurrence = 15, distant metastases = 6) after 0 to 91 months after completing treatment (mean and median values, respectively: 19.8 months ± 4.1 and 12.5 months).

### 3.2. Correlation between STING Immunoexpression and Epidemiological, Clinical and Histopathological Features

In the group of 77 patients, the mean and values of THS and SHS were, respectively: 61.8 ± 6.3 and 57.8% and 25.4 ± 4.5 and 22.0. The difference between mean values of THS and SHS was statistically significant (*p* = 0.025). On the basis of cut off point for THS and SHS, which was at the level of 10% (found by minimal *p*-value method), all tumors were stratified as those with tumor STING immunoexpression (TSI) and its lack as well as those with stromal STING immunoexpression (SSI) and its lack. TSI and SSI were both found in 55 patients (71.4%) (Table 1). The proportion of cancers with TSI was significantly (*p* = 0.005) higher in patients with regression of cancer disease (80.4%) than that in patients with progression of cancer. The distribution of cancers with STING immunoexpression/its lack was not significantly correlated with other epidemiological (patient’s age, gender, Karnofsky status, levels of smoking and drinking), clinical (T and N stage) and histopathological features (grade, keratinization status, active HPV16 infection). Tumors with a presence of SSI were significantly more often localized in the oral cavity than in the oropharynx (*p* = 0.000) and in cancers infected with HPV16 compared to cancers without active virus presence (*p* = 0.002). No other relation between the distribution of cancers with SSI or its lack and rest of the epidemiological, clinical and histopathological features was noticed.

### 3.3. Survival Analysis

In the series of 77 patients, OS and DFS were, respectively: 46.4% and 62.0%. In the univariate analysis, significantly better OS was noticed for female patients (*p* = 0.001), those without addiction to smoking (*p* = 0.042) and alcohol (*p* = 0.012), patients with lower T (*p* = 0.000) and N stages (*p* = 0.023), those having tumors without keratinization (*p* = 0.015) and those with active HPV16 infection (*p* = 0.023) (Table 2). TSI and SSI did not influence OS significantly. In the case of DFS, significantly higher survival was found for female patients (*p* = 0.023), patients with low levels of smoking (*p* = 0.044) and alcohol drinking (*p* = 0.013), those with a lower T stage (*p* = 0.001), those suffering from tumors with active HPV16 infection (*p* = 0.049) and patients with tumor STING immunoexpression (*p* = 0.031).

For patients with active HPV16 infection, Kaplain–Meier curves stratified by tumor and stromal STING immunoexpression are presented in Figure 3a–d.

In the multivariate analysis, all variables that were a significant influence on survival in the univariate analysis were included. This analysis revealed as independent prognostic factors gender (*p* = 0.014) and T stage (*p* = 0.006) for OS and T stage (*p* = 0.002) and tumor STING immunoexpression (*p* = 0.042) in the case of DFS (Table 3).

Separate analysis concerning the influence of TSI and SSI on patients’ survival was performed in the subgroups of patients with active HPV16 infection and without this infection. In the subgroup with active HPV16 infection, patients with tumors with THS had significantly better DFS (*p* = 0.047) than those without THS (Table 4). In this subgroup, TSH did not significantly influence OS, nor was SHS significantly correlated with OS and DFS. In the subgroup of patients without active HPV16 infection, THS and SHS also did not significantly influence patients’ survival. Due to low number of patients in these subgroups we did not perform multivariate analysis.

## 4. Discussion

In the present study, we have shown, according to our best knowledge for the first time, tumor STING immunoexpression as positive independent factor for disease-free survival of patients with oral cavity and oropharynx cancers (Table 3). Moreover, a significant relation between THS and DFS was also noticed in the subgroup of patients with active HPV16 infection, whereas in the subgroup without this infection THS did not significantly influence DFS (Table 4). In the whole group of patients, stromal STING immunoexpression significantly correlated with DFS in univariate analysis (Table 2); however, in multivariate analysis, it did not reach significance. Presented results are in line with those obtained by Luo et al. [12] in a group of 264 patients with HNSCC (32% HPV-positive). They have shown that higher STING immunoexpression in tumor parenchyma and tumor microenvironment are significantly correlated with improved OS. In the Cox multivariate regression model, they found that STING expression in tumor parenchyma remains an independent prognostic factor. After stratification of tumors by HPV status, STING expression in tumor parenchyma (similarly to our study) and in microenvironment (contrary to our study) correlated with patients’ survival in the HPV-positive group but not in the HPV-negative group. However, it should be noticed that these authors did not specify the localization of the analyzed HNSCC or the method of HPV assessment or the cut-off point obtained to distinguish higher/lower STING immunoexpression. Other authors reported similar results, although in a whole group of patients without stratification by HPV status. Zhu et al. [15], in a group of 327 OCSCC patients, have shown significantly better 10-year survival rate for patients with cGAS-STING high cluster as compared to patients with low cluster. Division into two clusters (high and low) was based on median value of enrichment score of six key genes of the GAS-STING pathway (cGAS, STING1, TBK1, IRF3, CCL5 and CXCL10). In turn, Hayman et al. [16], in a group of 52 patients with OPSCC, reported worse progression-free survival for patients with low tumor STING expression and independently stromal STING expression, both assessed by AQUA-based fluorescent analysis. There are also some papers concerning other types of tumors (non–small cell lung cancer, gastric, cervical or colorectal cancers) in which, similar to HNSCC, STING expression was related to better prognoses of patients [17,18,19,20].

There are a few hypotheses that can explain better prognoses for patients with STING expression/overexpression. One of them is associated with the influence of this pathway on the immune system. There is some evidence showing the influence of STING expression on the induction of type I IFN and the same maturation of dendritic cells, production of inflammatory cytokines and CD8+ cytotoxic T cells [7]. In vitro studies have also revealed that STING deficiency correlates with cancer incidence and that downregulation of this pathway induces resistance of cancer cells on the immune system [21]. It was shown that downregulation of the STING pathway is also correlated with a decrease in intratumoral CD8+ T cell infiltration and lower expression of some chemokines [22].

A second hypothesis explaining better prognoses for patients with STING expression is related to some data suggesting that cancer cells expressing STING are more susceptible to ionizing radiation (IR). Radiation was classically characterized as cytotoxic modality due to its ability to induce DNA lesions. However, there is increasing evidence showing that IR can act through the cGAS-STING pathway to alter cellular radiosensitivity and simulate the host immune system. Hayman et al. [16] have found that STING regulates a transcriptional program that controls the production of reactive oxygen species and that STING loss alters redox homeostasis to reduce DNA damage and at the same time cause radioresistance. Moreover, Liang et al. [23] proposed that radiation-induced STING activation suppressed immune response due to myeloid-derived suppressor cells infiltration, which results in tumor radioresistance. In turn, Deng et al. [24] reported that the cGAS-STING pathway is required for type I IFN induction after IR and that type I IFN response determinates the radiation-induced adoptive immune response. It should be also noticed that in our study, all patients were treated with IR; however, other authors do not provide information about the treatment that was used in the analyzed groups. cGAS–STING signaling and subsequent innate immune activation following DNA damage may function to alert the immune system to the presence of aberrant cellular phenotypes with potential for neoplastic transformation. Thus, radiation-induced DNA damage may permit exploitation of this innate immune-activating pathway via promoting cytosolic dsDNA accumulation and enable improved therapeutic efficacy against cancer. However, the order in which anticancer strategies are applied should be carefully considered. The use of radiotherapy prior to vascular disrupting agent/STING agonist administration has been shown to be more effective in murine melanoma growth inhibition than in either of the agents individually or in reverse combination [25]. However, contrary to the hypothesis about increased radiation response in cells with STING expression, Zheng et al. [26] reported that irradiation promotes tumor progression (induction of cancer lung metastasis) through activation of the cGAS-STING pathway in mesenchymal stromal cells. Therefore, the hypothesis about increased response to radiation in cells with STING expression requires further studies on the experimental and preclinical levels.

The third hypothesis concerning the explanation of better prognoses for patients with STING expression is related to, as observed by some authors, a correlation between STING expression and HPV infection. Experimental studies have revealed that viral oncoprotein E7 is responsible for blockading the cGAS-STING pathway in HPV16-positive OPSCC and that loss of E7 from these cells restored the cGAS-STING pathway [9,10]. Therefore, in the HPV16-positive cells, lack of STING expression should be expected. Contrary to these considerations, some authors have shown overexpression of STING in HPV-positive HNSCC tumor cells and in stromal cells compared to those without viral expression [11,12,13]. However, in the present study, we have shown, according to our best knowledge, for the first time significant differences in the distribution of tumors with STING expression in stroma between HPV16-positive and HPV16-negative tumors, though we did not obtain such difference when THS was analyzed (Table 1). These results suggest that one of the possible anticancer treatments consisting of the intratumoral administration of specific STING agonists would be more effective in HPV16-positive patients than in HPV16-negative ones. However, the molecular basis of these results are unknown. The fact that we noticed a significant correlation between HPV16 presence and a lack of stromal STING immunoexpression may suggest that HPV16 oncoproteins can influence stromal cells to suppress the immune response of the tumor microenvironment during the carcinogenesis process. On the other hand, in relation to contrary results concerning the relation between STING expression and HPV infection, attention must be also paid to methods used to assess viral presence. Most authors applied P16 expression as a surrogate marker of HPV infection [11,13]; however, in using this method, there is a risk of false positive results, because P16 overexpression is a result not only viral presence but also gene mutation or presence of DNA damage. Therefore, in our study, we decided to analyze active viral infection assessed on the basis of nested PCR, qPCR and P16 immunoexpression [14]. In turn, some authors suggest that the correlation between STING expression and HPV status may also depend on the subtype of virus because of different mechanisms of STING downregulation in the cases of HPV16 and HPV18 infection [27]. It was shown that HPV16 E7 protein exerts a direct inhibitory action on STING through its LCXCE motif [28] and HPV18 E7 through the LCXCE domain [12]. On the other hand, there are also some data suggesting HPV16 E7 modulates STING stability through the NOD-like receptor NLRX1 [28]. Therefore, to maintain the homogeneity of the analyzed group, we decided to limit our analysis to HPV16-positive tumors only and not take into account tumors infected with other type of virus.

## 5. Conclusions

The reported results indicate the potential prognostic value of tumor STING immunoexpression for patients with HPV16-associated head and neck cancers. Further analysis of these prognostic relationships should be confirmed in future translational studies. Experimental studies aimed at exploring related biological mechanisms are also needed to allow the field to fully leverage these findings for the benefit of patients.

## Figures and Tables

**Figure 1 biomedicines-10-02538-f001:**
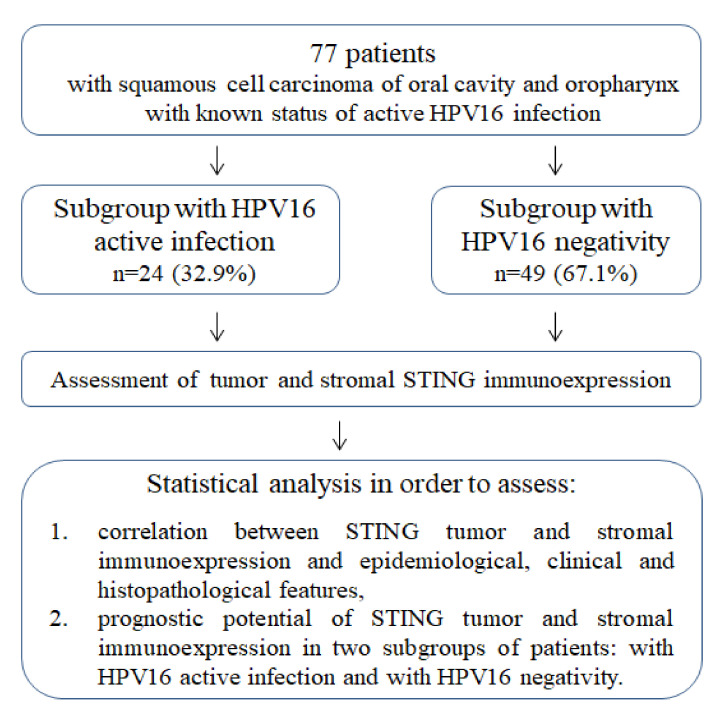
Study protocol diagram. Subgroups with HPV16 active infection and with HPV16 negativity were identified in earlier study [14].

**Figure 2 biomedicines-10-02538-f002:**
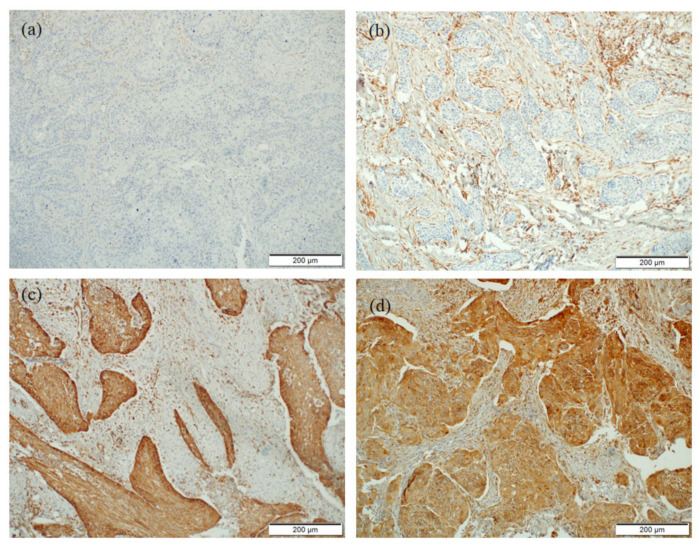
Representative microphotographs of STING immunostaining, (**a**) lack of STING immunostaining in tumor and in adjacent stromal tissues, (**b**) weak staining in adjacent stromal tissues, lack of staining in tumor area, (**c**) strong staining in tumor area, lack of staining in adjacent stromal tissues, (**d**) strong staining in tumor area and weak staining in adjacent stromal tissues.

**Figure 3 biomedicines-10-02538-f003:**
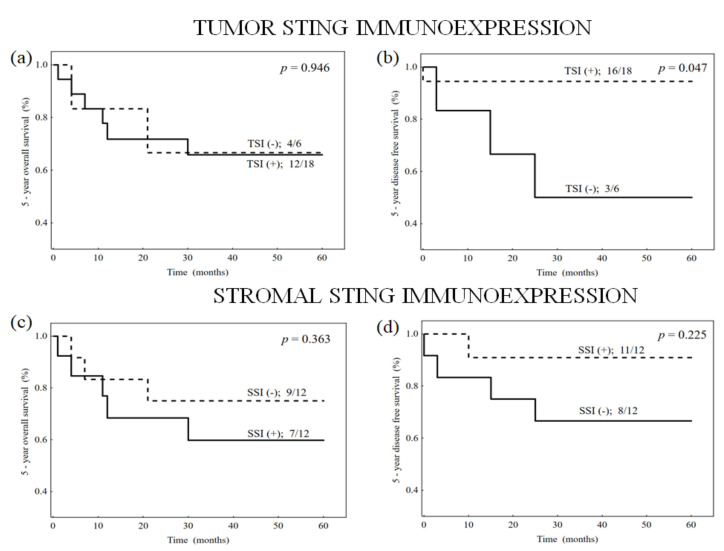
Kaplan–Meier curves concerning overall (**a**,**c**) and disease-free survival (**b**,**d**) stratified by tumor and stromal STING immunoexpression for patients with active HPV16 infection.

**Table 1 biomedicines-10-02538-t001:** Relation between tumor and stromal STING expression and epidemiological, histopathological and clinical features in the group of 77 squamous cell carcinomas of oral cavity and oropharynx.

Characteristics	All *n* (%) ^a^	Tumor STING Expression			Stromal STING Expression		
		Yes *n* (%) ^b^	No *n* (%)	*p* Level (χ^2^ Pearson)	Yes *n* (%) ^b^	No *n* (%)	*p* Level (χ^2^ Pearson)
All	77 (100.0)	55 (71.4)	22 (28.6)		55 (71.4)	22 (28.6)	
Age							
≤59 years	39 (50.6)	29 (74.4)	10 (25.6)		31 (79.5)	8 (20.5)	
>59 years	38 (49.4)	26 (68.4)	12 (31.6)	0.564	24 (63.2)	14 (36.8)	0.113
Gender							
Male	57 (74.0)	39 (68.4)	18 (31.6)		38 (66.7)	19 (33.3)	
Female	20 (26.0)	16 (80.0)	4 (20.0)	0.324	17 (85.0)	3 (15.0)	0.118
Status in the Karnofsky scale							
<80%	42 (54.6)	31 (73.8)	11 (26.2)		30 (71.4)	12 (28.6)	
≥80%	5 (45.4)	24 (68.6)	11 (31.4)	0.612	25 (71.4)	10 (28.6)	1.000
Localization							
Oral cavity	22 (28.6)	17 (23.7)	5 (77.3)		22 (1000.0)	0 (0.0)	
Oropharynx	55 (71.4)	38 (69.1)	17 (30.9)	0.473	33 (60.0)	22 (40.0)	**0.000**
The level of smoking—Brinkman index							
≤520	37 (48.1)	29 (78.4)	8 (21.6)		27 (73.0)	10 (27.0)	
>520	40 (51.9)	26 (65.0)	14 (35.0)	0.194	28 (70.0)	12 (30.0)	0.773
The level of drinking							
Low	34 (44.2)	25 (73.5)	9 (26.5)		26 (76.5)	8 (23.5)	
High	43 (55.8)	30 (69.8)	13 (30.2)	0.717	29 (67.4)	14 (32.6)	0.384
T stage							
2	15 (17.8)	11 (73.3)	4 (26.7)		11 (73.3)	4 (26.7)	
3	42 (49.2)	31 (73.8)	11 (26.2)		30 (71.4)	12 (28.6)	
4	20 (31.6)	13 (65.0)	7 (35.0)	0.760	14 (70.0)	6 (30.0)	0.977
N stage							
0	11 (14.3)	7 (63.6)	4 (36.4)		9 (81.8)	2 (18.2)	
1	16 (20.8)	11 (68.7)	5 (31.3)		10 (62.5)	6 (37.6)	
2	43 (55.8)	34 (79.1)	9 (20.9)		33 (76.7)	10 (23.3)	
3	7 (9.1)	3 (42.9)	4 (57.1)	0.220	3 (42.9)	4 (57.1)	0.203
Grade							
1	29 (37.7)	19 (65.5)	10 (34.5)		19 (65.5)	10 (34.5)	
2	41 (53.2)	32 (78.1)	9 (21.9)		30 (73.2)	11 (26.8)	
3	7 (9.1)	4 (57.1)	3 (42.9)	0.354	6 (85.7)	1 (14.3)	0.533
Keratinization							
Yes	44 (57.1)	32 (72.7)	12 (27.3)		34 (77.3)	10 (22.7)	
No	33 (42.9)	23 (69.7)	10 (30.3)	0.771	21 (63.6)	12 (36.4)	0.120
P16 immunoexpression							
Yes	24 (31.2)	19 (79.2)	5 (20.8)		12 (50.0)	12 (50.0)	
No	53 (68.8)	36 (67.9)	17 (32.1)	0.312	43 (81.1)	10 (18.9)	**0.005**
active HPV16 infection							
Yes	24 (32.9)	18 (75.0)	6 (25.0)		12 (50.0)	12 (50.0)	
No	49 (67.1)	34 (69.4)	15 (30.6)	0.619	41 (83.7)	8 (16.3)	**0.002**
Tumor STING expression							
Yes	55 (71.4)				46 (83.6)	9 (16.4)	
No	22 (28.6)				9 (40.9)	13 (59.1)	
Treatment							
Definitive CisPt-CRT or surgery + CisPt-CRT	28 (36.4)	20 (71.4)	8 (28.6)		18 (64.3)	10 (35.7)	
Definitive RT or surgery + RT	30 (39.0)	23 (76.7)	7 (23.3)		26 (86.7)	4 (13.3)	
Induction CT + definitive RT	19 (24.6)	12 (63.2)	7 (36.8)	0.594	11 (57.9)	8 (42.1)	0.074
Treatment outcome							
Regression of cancer disease	51 (66.2)	41 (80.4)	10 (19.6)		37 (72.5)	14 (27.5)	
Treatment failure	15 19.5)	7 (46.7)	8 (53.3)		11 (73.3)	4 (26.7)	
Local recurrence	6 (7.8)	2 (33.3)	4 (66.7)		3 (50.0)	3 (50.0)	
Distant metastases	5 (6.5)	5 (100.0)	0 (0.0)	**0.005**	4 (80.0)	1 (20.0)	0.662
Survival							
Alive at the last follow-up	37 (48.0)	27 (73.0)	10 (27.0)		25 (67.6)	12 (32.4)	
Death from cancer disease	22 (28.6)	12 (54.6)	10 (45.4)		15 (68.2)	7 (31.8)	
Death from other reasons	18 (23.6)	16 (88.9)	2 (11.1)	0.055	15 (83.3)	3 (16.7)	0.442

^a^ Column percentage, ^b^ Row percentage.

**Table 2 biomedicines-10-02538-t002:** Univariate Cox proportional hazard model for 5-year overall and disease-free survival of 77 patients with squamous cell carcinoma of oral cavity and oropharynx.

Characteristics	Overall Survival				Disease-Free Survival			
	Response *n* (%) *	HR	95% CI	Log-Rank *p*	Response *n* (%) *	HR	95% CI	Log-Rank *p*
Age:								
≤58 years ^a^	16/39 (41.0)	1.459			25/39 (64.1)	1.354		
>58 years	21/38 (55.3)	1.000	0.784–2.714	0.313	27/38 (71.0)	1.112	0.352–3.507	0.511
Gender								
Female	17/20 (85.0)	1.000			17/20 (85.0)	1.000		
Male	20/57 (35.1)	5.564	1.712–18.081	**0.001**	35/57 (61.4)	3.866	0.496–10.107	**0.023**
Status in the Karnofsky scale								
≤80%	18/42 (42.9)	1.682			27/42 (64.3)	1.205		
>80%	19/35 (54.3)	1.000	0.924–3.059	0.212	25/35 (71.4)	1.000	0.381–3.809	0.294
Localization								
Oral cavity	9/22 (40.9)	1.368			13/22 (59.1)	1.445		
Oropharynx	28/55 (50.9)	1.000	0.706–2.653	0.360	39/55 (70.9)	1.000	0.390–5.359	0.221
T stage								
1 + 2	34/57 (59.6)	1.000			44/57 (77.2)	1.000		
3 + 4	3/20 (15.0)	3.129	1.657–5.910	**0.000**	8/20 (40.0)	3.005	0.962–9.386	**0.001**
N stage								
0 + 1	18/27 (66.7)	1.000			19/27 (70.4)	1.000		
2 + 3	19/50 (30.0)	2.252	1.071–4.737	**0.023**	33/50 (66.0)	2.341	0.630–8.695	0.428
Grade								
1	16/29 (55.2)	1.000			21/29 (72.4)	1.000		
2	18/41 (43.9)	1.363	0.990–2.694		28/41 (68.3)	1.216	0,504–2.936	
3	3/7 (42.8)	1.428	1.311–3.609	0.648	3/7 (42.8)	1.455	0.796–2.659	0.556
Keratinization								
Yes	17/44 (38.6)	2.203			27/44 (61.4)	2.183		
No	20/33 (60.6)	1.000	1.134–4.280	**0.015**	25/33 (75.8)	1.000	0.938–5.080	0.058
The level of smoking—Brinkman index ^b^								
≤520	23/37 (62.2)	1.000			29/37 (78.4)	1.000		
>520	14/40 (35.0)	1.929	1.004–3.999	**0.042**	23/40 (57.5)	2.295	0.989–5.323	**0.044**
The level of drinking								
Low	23/34 (67.6)	1.000			28/34 (82.3)	1.000		
High	14/43 (32.6)	2.356	1.174–4.726	**0.012**	24/43 (55.8)	2.962	1.180–7.432	**0.013**
active HPV16 infection								
Present	16/24 (66.7)	1.000			15/24 (79.2)	1.000		
Absent	18/49 (36.7)	2.389	1.075–5.303	**0.023**	29/49 (59.1)	2.630	0.941–7.348	**0.049**
Tumor STING immunoexpression								
Yes	27/55 (49.1)	1.000			42/55 (76.4)	1.000		
No	10/22 (45.4)	1.047	0.532–2.060	0.891	10/22 (45.4)	2.282	1.040–5.004	**0.031**
Microenvironment STING immunoexpression								
Yes	25/55 (45.5)	1.376			38/55 (69.1)	1.000		
No	12/22 (54.5)	1.000	0.672–2.818	0.362	14/22 (63.6)	1.063	0.458–2.467	0.883
Treatment								
Definitive CRT or surgery + CRT	17/28 (60.7)	1.000			24/28 (85.7)	1.000		
Definitive RT or surgery + RT	14/30 (46.7)	1.484	0.688–3.201		19/30 (63.3)	2.015	1.127–3.602	
Induction CT + definitive RT	6/19 (31.6)	1.730	0.956–3240	0.356	9/19 (47.4)	2.865	0.910–9.006	0.059

Abbreviations: HR, hazard ratio; CI, confidence interval. * Row percentage; ^a^ Median values, ^b^ Number of cigarettes per day × years of smoking.

**Table 3 biomedicines-10-02538-t003:** Multivariate Cox proportional hazard model for disease-free survival of 77 patients with squamous cell carcinoma of the oral cavity and oropharynx.

Characteristics	HR	95% CI	*p*-Value ^a^
**Overall survival**			
Gender			
Female	1.000		
Male	4.501	1.363–14.865	0.014
T stage			
1 + 2	1.000		
3 + 4	2.466	1.293–4.701	0.006
**Disease-free survival**			
T stage			
1 + 2	1.000		
3 + 4	3.616	1.627–8.036	0.002
Tumor STING expression			
Yes	1.000		
No	3.912	1.915–9.443	0.042

Abbreviations: HR, hazard ratio; CI, confidence interval. ^a^
*p*-values were examined by the Cox proportional hazard model for multivariate survival analysis.

**Table 4 biomedicines-10-02538-t004:** The relations between tumor STING immunoexpression and stromal STING immunoexpression and overall survival or disease-free survival in the subgroups of patients with oral cavity and oropharynx cancers with active HPV16 infection (*n* = 24) and without this infection (*n* = 49).

	Overall Survival					Disease-Free Survival			
	Response *n* (%)		HR	95% CI	Log-Rank *p*	Response *n* (%)	HR	95% CI	Log-Rank *p*
**Active HPV16 infection**	Tumor STING immunoexpression								
	Yes	12/18 (66.7)	1.055	0.213–5.234	0.946	17/18 (94.4)	1.000	0.701–15.217	**0.047**
	No	4/6 (66.7)	1.000			3/6 (50.0)	4.206		
	Stromal STING immunoexpression								
	Yes	7/12 (58.3)	1.916	0.547–8.035	0.363	11/12 (91.7)	1.000	0.397–12.025	0.225
	No	9/12 (75.0)	1.000			8/12 (66.7)	3.566		
**Lack of active HPV16 infection**	Tumor STING immunoexpression								
	Yes	13/34 (38.2)	1.000	0.503–2.275	0.853	24/35 (68.6)	1.000	1.040–5.004	0.130
	No	5/15 (33.3)	1.070			6/15 (40.0)	2.282		
	Stromal STING immunoexpression								
	Yes	16/41 (39.0)	1.000	0.484–2.880	0.706	25/41 (61.0)	1.000	0.365–3.278	0.869
	No	2/8 (25.0)	1.180			4/8 (50.0)	1.093		

## Data Availability

All data generated or analyzed during this study are included in this article. Further enquiries can be directed to the corresponding author.

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
