# Peer review of "Prognostic Significance of STING Immunoexpression in Relation to HPV16 Infection in Patients with Squamous Cell Carcinomas of Oral Cavity and Oropharynx"

_biomedicines, 2022, doi:10.3390/biomedicines10102538_

Round 1
Reviewer 1 Report
Abstract:
Line 28: “The study was 27 performed in the group of 87 patients with OCSCC and OPSCC, for which in our earlier study 28 active HPV16 infection was assessed.”
Active infection, is a not considered to be a feature of HPV+ HNSCC, therefore more support for this statement is needed, or as it is not particularly critical, probably better to just avoice the issue.
I suggest something like “The study was performed in the group of 87 patients with OCSCC and OPSCC, for which in our earlier study assessed HPV16 status with XXXX technique.”
Line 29: Throughout the paper I would substitute HPV positivity in place of active infection, unless you can justify scientifically that the virus is “alive” or infectious, which would be difficult.
Line 32: I suggest, “In this subgroup, TSH did not significantly influence OS, AND SHS did 32 not significantly correlated with OS OR DFS.”
Introduction:
Line 40: Recently, human papillomavirus (HPV) infection is the cause of a growing number 40 of squamous cell carcinomas of head and neck (HNSCCs), especially within oropharynx (OPSCC) and oral cavity (OCSCC) [1].
Does this reference really support the assertion that some oral cavity tumors are HPV associated? This is a contentious issue; likely these are mis-classified tumors of the tongue base. Might just be easier to avoid this issue.
I suggest “Recently, human papillomavirus (HPV) infection is the cause of a growing number of squamous cell carcinomas of head and neck (HNSCCs), especially within oropharynx (OPSCC) [1].”
Line 76: “In our earlier study, among 87 patients with OCSCC and OPSCC we found HPV16 76 transcriptionally active infection (P16 overexpression and positivity of HPV DNA in 77 quantitative polymerase chain reaction - qPCR) in, respectively 16.0 and 37.1%, in lar- 78 yngeal and hypopharyngeal cancers, these percentages were significantly lower [14].”
I suggest something like: “In our earlier study, among patients with HNSCC we found evidence of HPV16 positivity (P16 overexpression and positivity for HPV DNA by quantitative polymerase chain reaction - qPCR) in only 16.0 and 37.1%, in laryngeal and hypopharyngeal cancers, respectively. Considering these low percentages, we excluded laryngeal and hypopharyngeal cancers from our present study [14].”
Results:
Table 1-4. “HPV16 active infection”
I suggest changing this language to HPV16 positivity, unless you can provide further evidence of infection or infectious virus.
Line 188: I think you mean “TSI and SSI were BOTH found in 55 patients (71.4%) (71.4%) (Table 1).”
Overall, it is atypical to show no survival curves in a manuscript which is ultimately about survival analysis. The intro focuses on HPV+ de-escalation. I think showing the KM plots for the HPV+ stratified by tumor STING immunoexpression is needed.
Discussion:
Line 249, I suggest: After stratification of tumors by HPV status, 249 STING expression in tumor parenchyma (similarly to our study) and in microenvironment 250 (contrary to our study) correlated with patient’s survival in HPV positive group, but not in HPV 251 negative group.
Line 260: In turn, Hayman et al. [16] in the group of 52 260 patients with OPSCC, reported worse progression free survival for patients with low 261 tumor STING expression and independently stromal STING expression assessed by 262 AQUA-based fluorescent analysis.
Sentence needs to be clarified, not sure about the meaning of the second half of the sentence.
Line 311: In 311 line with these considerations some authors have shown overexpression of STING in 312 HPV positive HNSCC tumor cells and in stromal cells compared to those without viral 313 expression [11-13].
Isn’t this contrary to, no “in line” with the findings outlined in the preceding sentences? Please clarify.
Line 329: “Therefore, in our study we decided to analyze active viral infection assessed on the basis of nested PCR, qPCR and P16 immunoexpression [14].”
Again, P16 and qPCR can’t show “active infection”, tumors with integrated viral genomic material alone can have these features. Please re-word this and other such instances.
Conclusions 331
The results indicate prognostic potential of tumor STING immunoexpression for 332 patients with HPV16 active infection in oral cavity and oropharynx cancers. These results 333 are important suggestion that further analysis concerning prognostic potential of this 334 protein should be performed separately in patients with active HPV infection and with- 335 out virus presence. Experimental studies to explain biological mechanism of this finding 336 are also expected.
I suggest:
“The reported results indicate potential prognostic value of tumor STING immunoexpression for patients with HPV16 associated head and neck cancers. Further analysis of these prognostic relationships should be confirmed in future translational studies. Experimental studies aimed at exploring related biological mechanisms are also needed, to allow the field to fully leverage these findings for the benefit of patients.”
Reviewer 2 Report
Biesaga et al. have identified the prognostic significance of STING expression in HPV-positive oral cavity and oropharyngeal cancers. This work is also in line with the previous work of Shamseddine et al. published in Cancer Discovery (33990345). The reviewer is curious to know whether this correlation of STING expression in head and neck cancer patients is only with HPV16 infection or with other HPV subtypes. The reviewer suggests the authors clarify different HPV subtypes in the discussion part in correlation with STING expression.
Reviewer 3 Report
- the association between head and neck squamous cell carcinoma (HNSCC) and infection with different human papillomavirus virus (HPV) subtypes, including analysis of promoter methylation of several genes (APC, CDKN2A, MGMT, CDH1 and TIMP3) and the correlation with their mRNA expression in tumours and surgical margins is relevant, please discuss and cite doi:10.1016/j.micpath.2020.104692 and doi:10.1099/jmm.0.000898
- functional swallowing reserve and phonatory function are highly dependent on organ preservation and the patient's initial state, allow for more aggressive but resolving treatments versus medical therapy, which instead does not allow eradication of the disease., please discuss and cite doi:10.1007/s00405-016-4177-0
-the prognosis of hpv+ patients is strongly influenced by the stage at diagnosis as well as the treatment administered, which also affects their quality of life and side effects., doi:10.1016/j.anl.2021.05.007
- please add consort model diagram to improve the description of the study protocol.
